# Learning Efficient Robotic Garment Manipulation with Standardization

**Changshi Zhou** [1 2 3]  **Feng Luan** [1 2 3]  **Jiarui Hu** [2 3 4]  **Shaoqiang Meng** [1 2 3]  **Zhipeng Wang** [2 3 4]  **Yanchao Dong** [2 3 4]
**Yanmin Zhou** [2 3 4]  **Bin He** [2 3 4]

## Abstract

Garment manipulation is a significant challenge for robots due to the complex dynamics and potential self-occlusion of garments. Most existing methods of efficient garment unfolding overlook the crucial role of standardization of flattened garments, which could significantly simplify downstream tasks like folding, ironing, and packing. This paper presents APS-Net, a novel approach to garment manipulation that combines unfolding and standardization in a unified framework. APS-Net employs a dual-arm, multi-primitive policy with dynamic fling to quickly unfold crumpled garments and pick-and-place(p&p) for precise alignment. The purpose of garment standardization during unfolding involves not only maximizing surface coverage but also aligning the garment's shape and orientation to predefined requirements. To guide effective robot learning, we introduce a novel factorized reward function for standardization, which incorporates garment coverage (Cov), keypoint distance (KD), and intersection-over-union (IoU) metrics. Additionally, we introduce a spatial action mask and an Action Optimized Module to improve unfolding efficiency by selecting actions and operation points effectively. In simulation, APS-Net outperforms state-of-the-art methods for long sleeves, achieving 3.9% better coverage, 5.2% higher IoU, and a 0.14 decrease in KD (7.09% relative reduction). Real-world folding tasks further demonstrate that standardization simplifies the folding process. Project page: https://hellohaia.github.io/APS/.

---

[1]Shanghai Research Institute for Intelligent Autonomous Systems, Tongji University, Shanghai, China [2]The National Key Laboratory of Autonomous Intelligent Unmanned Systems, Tongji University, Shanghai 201210, China [3]The Frontiers Science Center for Intelligent Autonomous Systems, Shanghai 201210, China [4]College of Electronics and Information Engineering, Tongji University, Shanghai 201804, China. Correspondence to: Yanmin Zhou <yanmin.zhou@tongji.edu.cn>.

*Proceedings of the 42nd International Conference on Machine Learning*, Vancouver, Canada. PMLR 267, 2025. Copyright 2025 by the author(s).

## 1. Introduction

Garment manipulation(Zhang & Demiris, 2022; Hoque et al., 2022) presents a critical challenge in robotics, with applications ranging from household assistance(Xiao et al.) and healthcare(Moglia et al., 2024; Wang et al., 2025) to industrial automation(Arents & Greitans, 2022). Unlike rigid objects(Yang et al., 2024a), garments are deformable(Jing et al., 2023), exhibit complex dynamics(Lin et al., 2022; Zhou et al., 2025b), and are prone to self-occlusion(Huang et al., 2022), which makes them particularly difficult to manipulate. These challenges are amplified when tasks such as folding(Xue et al., 2023), ironing(Canberk et al., 2023), sorting(Jing et al., 2024), and packing(Chen et al., 2024) require precise control over the garment's shape, orientation, and coverage.

While significant progress has been made in garment unfolding(Zhu et al., 2023), existing methods—especially those based on single-arm systems(Salhotra et al., 2023; Chen et al., 2023; Yang et al., 2024b)—often require many interactions and long execution times. Dual-arm systems(Ha & Song, 2022; He et al., 2024) that leverage dynamic fling actions have been introduced to speed up the unfolding process, but they focus primarily on maximizing surface coverage, neglecting the crucial task of garment standardization. Standardization—aligning the garment's shape, ensuring consistent orientation(Gu et al., 2024b), and achieving maximum surface coverage—is essential for successful downstream tasks, such as folding and ironing. Without standardization, further manipulation becomes increasingly difficult, leading to suboptimal outcomes.

To address these challenges, we propose a novel approach for garment manipulation, called the Action-Primitive Selector Network (APS-Net), which learns to predict spatial action maps—actions defined on a pixel grid—using overhead RGB-D images. APS-Net integrates both garment unfolding and standardization into a unified framework. By employing a dual-arm, multi-primitive policy, APS-Net intelligently selects between two key actions: dynamic fling and p&p (see Figure 1). The dynamic fling action is used to rapidly unfold crumpled garments, while p&p ensures fine-tuned alignment and standardization, preparing the garment for downstream tasks such as folding.

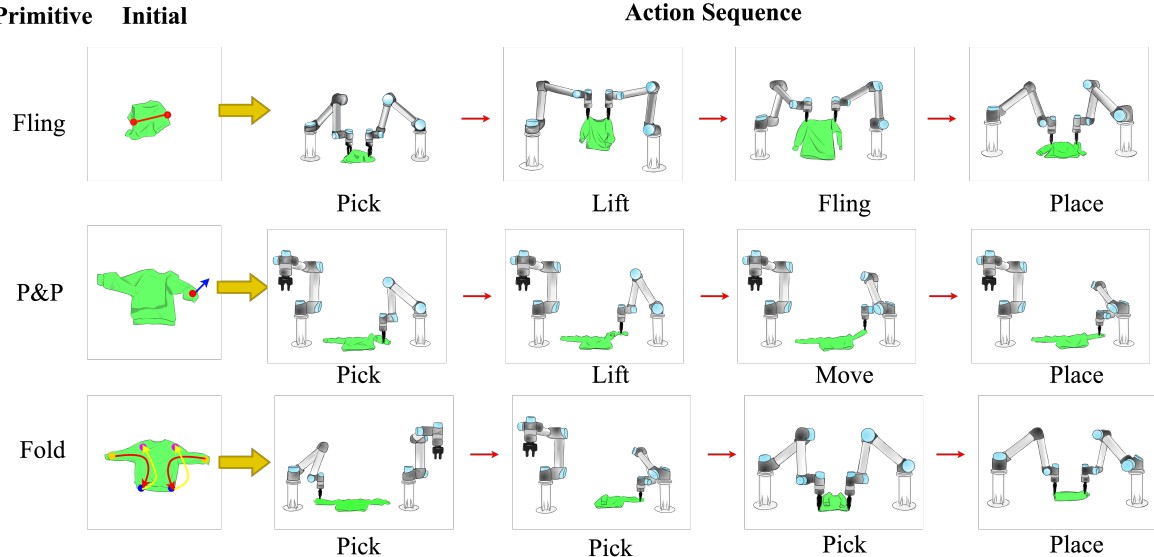

*Figure 1.* **Action Primitives.** The fling action involves the robot grasping two predefined points on the garment, lifting and stretching it, then performing a controlled fling motion forward and backward while gradually lowering it onto the workspace. The p&p action involves the robot grasping the garment at a specified pick point, lifting it, moving it to a target place point, and releasing it. In the fold action, keypoint detection selects the pick and place points, and both arms execute heuristic folding.

To ensure effective standardization, we introduce a novel factorized reward function for each action primitive. This function incorporates garment coverage, keypoint distance, and IoU metrics both before and after each action. This ensures that once unfolded, the garment achieves maximum coverage, maintains consistent orientation, and aligns with its goal shape, while preserving visible keypoints, which are essential for downstream folding tasks. Further, to address the issue of invalid action selections—such as targeting areas outside the garment or regions prone to arm collisions—we introduce a spatial action mask that filters out actions directed at void regions. Additionally, we present the action optimized module, which enhances unfolding efficiency by focusing on keypoints, particularly the shoulder regions, during the dynamic fling process.

To validate the effectiveness of garment standardization in downstream tasks like folding, we combine garment keypoint detection with heuristic folding methods. Additionally, we collect real images of various garment types and create a dedicated dataset for garment folding keypoint detection. Experimental results demonstrate that our method successfully unfolds arbitrarily crumpled garment configurations into standardized states, simplifying the downstream task.

The main contributions of this paper are as follows:

- We propose APS-Net, a self-supervised learning network that intelligently selects between fling and p&p actions. It effectively unfolds garments from arbitrary crumpled states into standardized configurations, while preserving clear visible keypoints for downstream tasks.

- We introduce a factorized reward function that combines garment coverage, IoU, and keypoints to guide APS-Net training.

- We design a spatial action mask to filter out invalid actions and introduce an action optimized module that enhances unfolding efficiency by targeting shoulder keypoints during dynamic fling.

- We conduct extensive real-world experiments on garments from three categories—long sleeve, jumpsuits, and skirts—demonstrating the effectiveness of APS-Net in both garment standardization and subsequent folding tasks using a dual-UR5 robot.

## 2. Related work

### 2.1. Garment unfolding

With the advancement of deep learning (Yang et al., 2025a; Bi et al., 2025b; Xin et al., 2024; Li et al., 2024; Yang et al., 2025b; Liu et al., 2022), large language models (LLMs) (Jiang et al., 2025; Tan et al., 2025; Bi et al., 2025a), and reinforcement learning models (Zhou et al., 2025a; Tang et al., 2025; Havrilla et al., 2024), an increasing number of studies have begun to address cloth manipulation within the

reinforcement learning framework. Among these, garment unfolding (Wu et al., 2024; Raval et al., 2024) remains a particularly challenging task, requiring intricate manipulation to transform crumpled garments into flat, standardized states. Early approaches relied on hand-crafted heuristics (Seita et al., 2020; Sun et al., 2014) or single-arm manipulation strategies (Qian et al., 2020; Lee et al., 2021), often requiring multiple interactions to achieve successful unfolding. More recent methods, such as those by Zhu et al. (2023) and Chen et al. (2023), employ supervised learning techniques like semantic segmentation and grasping pose estimation to improve unfolding performance. Wu et al. (2023) further introduced topological visual correspondences to handle garments with varying deformations. However, these methods still rely heavily on large labeled datasets and struggle when dealing with heavily wrinkled or deformed garments.

Reinforcement learning (RL) provides a promising alternative by enabling robots to optimize unfolding strategies through direct interaction with the environment. Chen et al. (2022) introduced high-velocity dynamic fling actions, but performance plateaus once a surface coverage threshold is reached. Avigal et al. (2022) and He et al. (2024) combined dynamic fling with static pick-and-place actions, but primarily focused on coverage while neglecting standardized alignment. Canberk et al. (2023) explored both single-arm and dual-arm manipulation, though their approach suffers from a significant sim-to-real gap that limits applicability in real-world scenarios. To address these challenges, we propose APS-Net, which ensures efficient garment unfolding with standardized alignment. To reduce the sim-to-real gap, APS-Net uses RGBD input for spatial accuracy, simulates cloth with variable mass and stiffness, and adds procedural wrinkles to enhance realism.

### 2.2. Garment folding

Garment folding(Gu et al., 2024a) is crucial in applications such as hospitals, homes, and warehouses, where efficient fabric manipulation is essential. Early methods relied on traditional image processing to identify garment features(Maitin-Shepard et al., 2010; Doumanoglou et al., 2016; Stria et al., 2014; Ma et al., 2024), but their strong assumptions about garment types and textures limited generalization to real-world scenarios. More recent approaches leverage goal-conditioned policies(Zhou et al., 2024; Chane-Sane et al., 2021; Kim et al., 2024) using reinforcement learning, self-supervised learning, and imitation learning. For example, Mo et al. (2022) introduced a space-time attention mechanism for multi-step folding, while Weng et al. (2022) used optical flow techniques to improve performance. However, these methods still depend on predefined goal states, which are often unavailable for novel instances. Recent research has focused on learning-based methods to detect key garment features, such as corners and edges, to

reduce task complexity and accommodate diverse garment types. Lips et al. (2024) and Canberk et al. (2023) used synthetic data for training keypoint detection models, achieving effective results, but struggled with the sim-to-real gap in real-world. To address these challenges, we collect a diverse dataset of 10,000 garment states with a coverage rate above 0.8 during ASP-Net training. We then train a keypoint detection model with DeepLabv3(Chen et al., 2017) and fine-tune it using 1,100 real-world images, improving robustness and adaptability in practical scenarios.

## 3. Methods

### 3.1. Problem Formulation

Standardized garment manipulation involves a sequence of actions that transform a crumpled garment into a standardized, aligned state, followed by folding it into a desired configuration. At each time step $t$, given an RGB-D image $o_t \in \mathbb{R}^{W \times H \times 4}$, an action $a_t$ of type $m \in \mathcal{M}$ is selected based on the policy $\pi_\theta$, where $\mathcal{M}$ is a discrete set of predefined action primitives (See Figure 1). The policy is represented as follows:

$$\pi_\theta(o_t) \to a_t \qquad (1)$$

The chosen action $a_t$ is then applied to the garment, resulting in a new state $o_{t+1}$, determined by the state transition $\mathcal{T}$:

$$o_{t+1} \to \mathcal{T}(o_t, a_t) \qquad (2)$$

This process continues until the garment reaches a sufficiently smooth state $o_{\text{smooth}}$. After which keypoint detection identifies features for final folding:

$$K_o = K_{\text{detect}}(o_{\text{smooth}}) \qquad (3)$$

The garment is then folded based on the detected keypoints using the fold primitive:

$$o_{\text{folded}} = f_{\text{fold}}(o_{\text{smooth}}, K_o) \qquad (4)$$

During this process, we formulate the unfolding process as a self-supervised learning problem, where the policy $\pi_\theta$ is learned. The folding process is treated as a supervised learning problem, where $K_{\text{detect}}$ is learned for keypoint detection.

### 3.2. Action-Primitive Selector Network (APS-Net)

Coarse-grain dynamic dual-arm flings can efficiently unfold garments from crumpled states, but are insufficient for the fine-grained adjustments required to achieve standardized alignment. To address this, we introduce the Action-Primitive Selector Network (APS-Net) (See Figure 2), which intelligently selects the optimal manipulation action—fling or p&p—for efficient and accurate garment unfolding. Garment unfolding requires precise grasping point

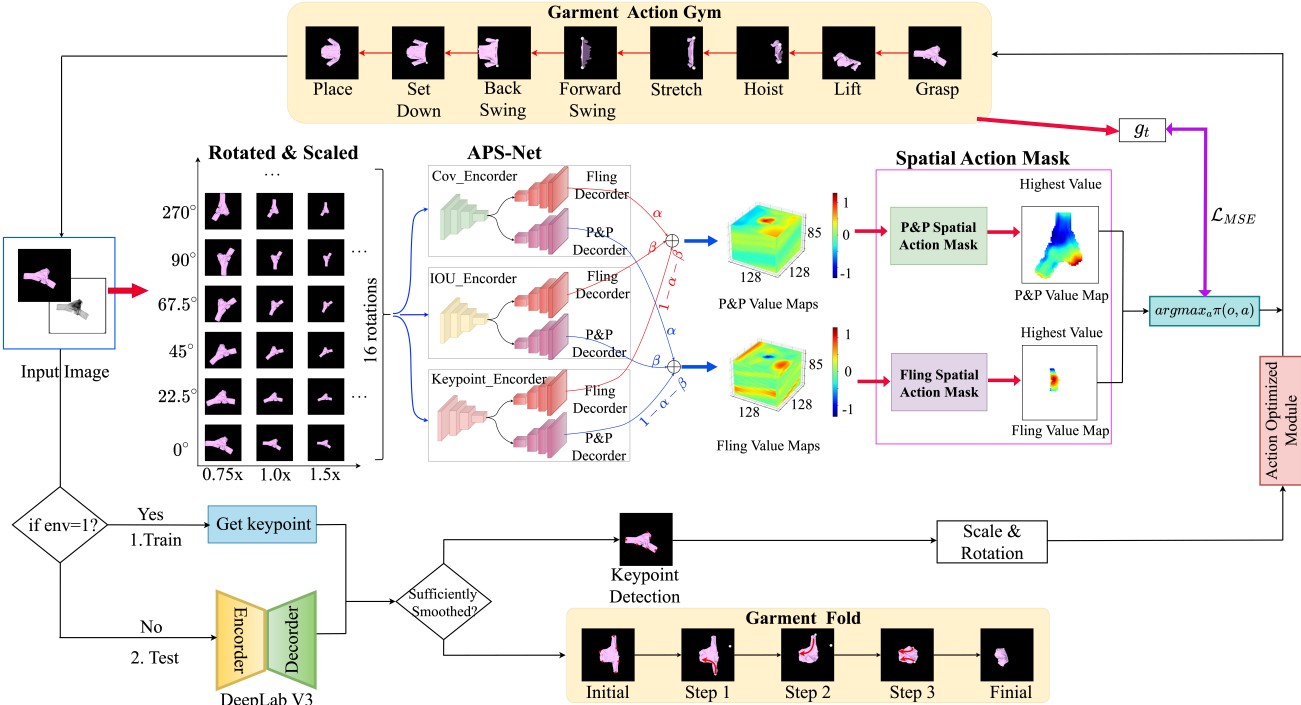

Figure 2. **Approach overview.** APS-Net takes a batch of rotated and scaled RGBD images as input and uses three encoders, each producing a pair of decoders — the fling and p&p decoders — which are weighted to generate the corresponding Spatial Action Map. A Spatial Action Mask is then applied to filter out invalid actions. The valid primitive batches are concatenated, and the action to be executed is parameterized by the maximum value pixel. Once the garment is sufficiently flattened, a keypoint detection-based method is employed to perform the folding.

selection for effective manipulation. For dual-arm fling, directly predicting both grasp points is challenging due to collision avoidance. Instead, we reframe the problem by predicting the midpoint (x, y), angle $\theta$ (rotation), and grasp width $w$, with collision avoidance achieved by adjusting $w$. For single-arm p&p, collision avoidance is not required. However, to maintain consistency with dual-arm fling and provide a unified output from the same network, we define (x, y) as the pick point, while $\theta$ and $w$ represent the rotation and length between the pick and place points. Therefore, we generalize the two cases as solving for $\langle x, y, \theta, w \rangle$.

Even so, the naive approach of directly predicting $\langle x, y, \theta, w \rangle$ fails to account for the invariance of the optimal grasp under the garment's physical transformations. To address this, we apply a series of transformations—rotation and scaling—to the observation $o_t$, generating a set of transformed images $T_n$ to ensure that the grasp points remain consistent across identical garment configurations. APS-Net then processes these transformed garment images in batches to generate spatial action maps representing the desirability of performing each action at each pixel of the garment. For each action primitive $m \in \{\text{fling, p\&p}\}$, APS-Net computes the maximum desirability across the set of transformed

value maps $\{V_{(m,1)}, V_{(m,2)}, \dots, V_{(m,n)}\}$ as follows:

$$\hat{V}_i^m = \max(V_i^m) \quad \forall i \in \{1, 2, \dots, n\} \tag{5}$$

Here, $\hat{V}_i^m$ represents the maximum desirability value found in the $i$-th transformed value map for action $m$.

The final action primitive is determined by comparing the maximum desirability values from the spatial action maps:

$$m = \begin{cases} \text{fling} & \text{if } \hat{V}_i^{\text{fling}} > \hat{V}_i^{\text{pick}} \\ \text{p\&p} & \text{otherwise} \end{cases} \tag{6}$$

Subsequently, the final value $V_{\text{final}}^{\max}$ is selected based on the chosen action primitive:

$$V_{\text{final}}^{\max} = \begin{cases} V_{(\text{fling},i)} & \text{if } m = \text{fling} \\ V_{(\text{pick},i)} & \text{if } m = \text{p\&p} \end{cases} \tag{7}$$

The index $i$ is then obtained from the APS-Net output as follows:

$$\langle x, y, i \rangle \leftarrow \text{APSNet}(T_n) = \arg\max\left(V_{f_{\max}}^{\max}\right) \tag{8}$$

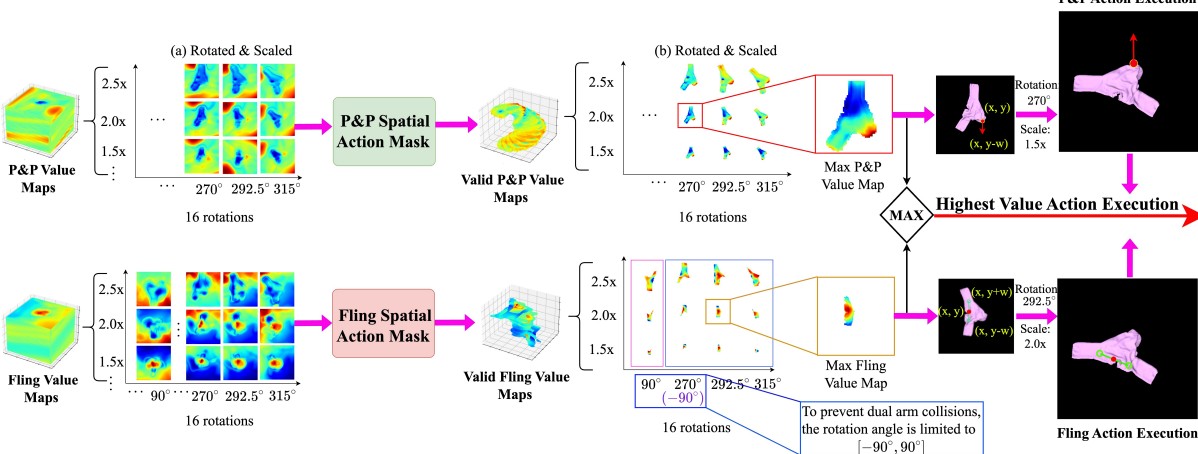

*Figure 3.* **Spatial Action Mask.** The series of smaller maps in (a) represents different slices of the spatial action maps for various primitives, each corresponding to a layer at a different scale and rotation. The masks applied to each layer serve to filter out invalid actions — those that would cause the robot's end-effector to collide or extend beyond the garment.

Once $i$ is determined, the corresponding width $w$ and rotation angle $\theta$ are retrieved from the $i$-th entry of the proposal set $T_n$:

$$w, \theta = T_n[i] \tag{9}$$

Based on the above reasoning, we obtain $\langle x, y, \theta, w \rangle$. Note that $\theta$ is a discrete variable. As described in Appendix E, $\theta$ is quantized into 16 discrete rotation angles to cover the full 360° range.

The grasp points $a_t$ are defined based on the selected action primitive:

$$a_t = \begin{cases} \{(x, y + w), (x, y - w)\} & \text{if } m = \text{fling} \\ \{(x, y), (x, y - w)\} & \text{if } m = \text{p\&p} \end{cases} \tag{10}$$

According to the definition (see Figure 1), for single-arm p&p, both the pick and place points need to be predicted. As indicated in Equation (10), when $m = \text{p\&p}$, the pick point is $(x, y)$, and the place point is $(x, y - w)$.

For dual-arm fling, only the two pick points are predicted, while the place points follow a predefined fling trajectory, as described in Equation (11). Specifically, when $m = \text{fling}$, the dual-arm pick points are $(x, y + w)$ and $(x, y - w)$. During execution, both arms follow the trajectory through lift, swing, and placement phases, dynamically adjusting acceleration and velocity at each stage to perform the fling motion.

$$\begin{aligned} \text{fling} = [(0, 0, h_l) &\rightarrow (0, f_m, h_l) \rightarrow \\ (0, f_m - b_m, h_l) &\rightarrow (0, f_m, h_p)] \end{aligned} \tag{11}$$

Here, $h_l$ is the lift height, $f_m$ denotes the forward swing distance, $b_m$ is the backward swing offset, and $h_p$ is the final placement height.

To prevent collisions during dual-arm fling operations, a constraint is imposed to ensure that the left grasp point $(L)$ remains to the left of the right grasp point $(R)$. This is enforced by limiting the rotation angle $\theta$ to the range $[-90°, 90°]$, ensuring safe and coordinated movements.

### 3.3. Factorized Reward Function

In this work, we propose a novel factorized reward function to train the policy to manipulate the garment into standardized unfolding. APS-Net consists of three encoders, each computing a distinct reward: coverage reward $R_C$, Intersection over Union (IoU) reward $R_I$, and keypoint reward $R_K$. Each encoder is paired with two decoder heads—one for each action primitive, fling and p&p. These rewards are computed separately for each action primitive and then combined into a weighted sum to guide network training.

To balance the contributions of these reward functions, a weighted sum is applied for each action primitive:

$$R_{CIK} = \alpha \cdot R_C + \beta \cdot R_I + (1 - \alpha - \beta) \cdot R_K \tag{12}$$

where $\alpha, \beta \in (0, 1)$ are hyperparameters. By tuning $\alpha$ and $\beta$, this reward mechanism enables APS-Net to optimize fabric coverage, alignment, and keypoint positioning, achieving a standardized garment configuration critical for subsequent folding tasks. See Appendix A for more details.

### 3.4. Spatial Action Mask

Directly sampling actions on the spatial policy maps can lead to invalid actions, such as empty actions in non-garment areas or double-arm collisions. To address this, we introduce the Spatial Action Mask (SAM), which filters out infeasible actions, ensuring that the robot only considers feasible

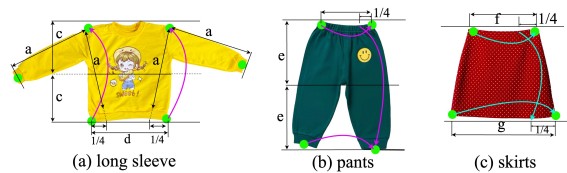

(a) long sleeve      (b) pants      (c) skirts

*Figure 4.* The folding rules for long sleeves, pants, and skirts.

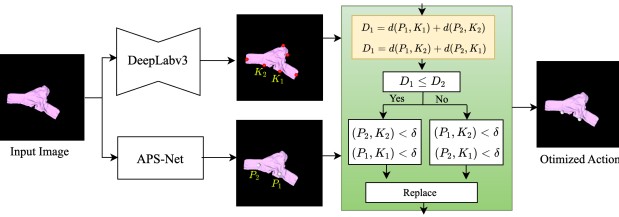

*Figure 5.* **Action Optimized Module.** Optimizes the alignment of garment shoulder keypoints for efficient flattening, based on the predicted shoulder points from APSNet.

actions within the valid workspace (See Figure 3). Specifically, we define both workspace and garment masks for each action primitive, and the valid action space is determined by their intersection. See Appendix B for more details.

### 3.5. Keypoint-Based Strategy for Garment Folding

Heuristics are highly interpretable and easy to define, making them effective for garment folding. In this approach, a DeepLabv3 detector(Chen et al., 2017) is trained for each garment class, and keypoints from the detected garments are depth-projected into 3D space, then transformed into the workspace frame of reference. By representing garment as a set keypoints, we sidestep its infinite DoF by using a few meaningful keypoints as the representation, which makes it simpler to define heuristics over them.

For long-sleeve garments, we define six key points: two at the sleeves, two at the shoulders, and two at the waist. First, fold the sleeves along the folding lines (Figure 4(a)), then fold the garment in half along the central line. For pants and skirts (Figure 4(b) and 4(c)), we define four key points: two at the waist and two at the bottom. Begin by folding along the waist and bottom lines, then fold 3/4 of the waist down toward the bottom key point to complete the fold.

### 3.6. Action Optimized Module

When performing a fling action to flatten a garment, keypoints on the shoulders are typically prioritized for efficient flattening. The Action Optimized Module (AOM) refines the APSNet-predicted points $P_1$ and $P_2$ by aligning them

with the garment's shoulder keypoints $K_1$ and $K_2$, if they are sufficiently close, ensuring quick flattening(See Figure 5). The module calculates the Euclidean distance between each predicted point and the corresponding keypoint:

$$d(P_i, K_j) = \sqrt{(P_{i,x} - K_{j,x})^2 + (P_{i,y} - K_{j,y})^2} \quad (13)$$

where $i, j \in \{1, 2\}$. Two possible matching combinations are evaluated, with $p_1$ paired with $k_1$ and $p_2$ paired with $k_2$, which results in the following:

$$D_1 = d(P_1, K_1) + d(P_2, K_2) \quad (14)$$

and pairing $p_1$ with $k_2$ and $p_2$ with $k_1$, yielding:

$$D_2 = d(P_1, K_2) + d(P_2, K_1) \quad (15)$$

The module selects the pair with the minimum total distance, $\min(D_1, D_2)$. If both distances in the selected pair satisfy $d(P_i, K_j) < \delta$, where $\delta = 5$, the predicted points $P_1$ and $P_2$ are replaced by $K_1$ and $K_2$. Otherwise, they are retained.

## 4. EXPERIMENTS

### 4.1. Tasks and Metrics

**Intersection over Union (IoU):** Measures the quality of Garment Unfold by comparing the manipulated Garment's mask with the target mask.

**Coverage (Cov):** Compares initial and achieved Garment coverage to the maximum possible area.

**Success Rate (SR):** A task is considered successful if the robot first smooths the crumpled garment and then correctly executes the folding operation according to the specified fold primitive.

*In simulation:* The cloth is modeled as a particle-based grid, where ground-truth particle positions are available. We compute the mean distance between particles in the achieved and target cloth states. A trial is deemed successful if the average particle distance error is less than 0.03.

*In the real world:* We use the Intersection over Union (IoU) between the predicted cloth mask and that of a human demonstrator to evaluate performance. If the IoU exceeds 0.8, the folding is considered successful.

**Keypoint distance (KD):** Represents the distance between key feature points of the garment.

These metrics are applied to three representative garment manipulation tasks: unfolding and folding long sleeves, skirts, and pants. In all tables, ↓ indicates that lower is better while ↑ indicates that larger values are preferred.

### 4.2. Baselines

We compare our method with several baselines, including quasi-static, dynamic, and state-of-the-art (SOTA) ap-

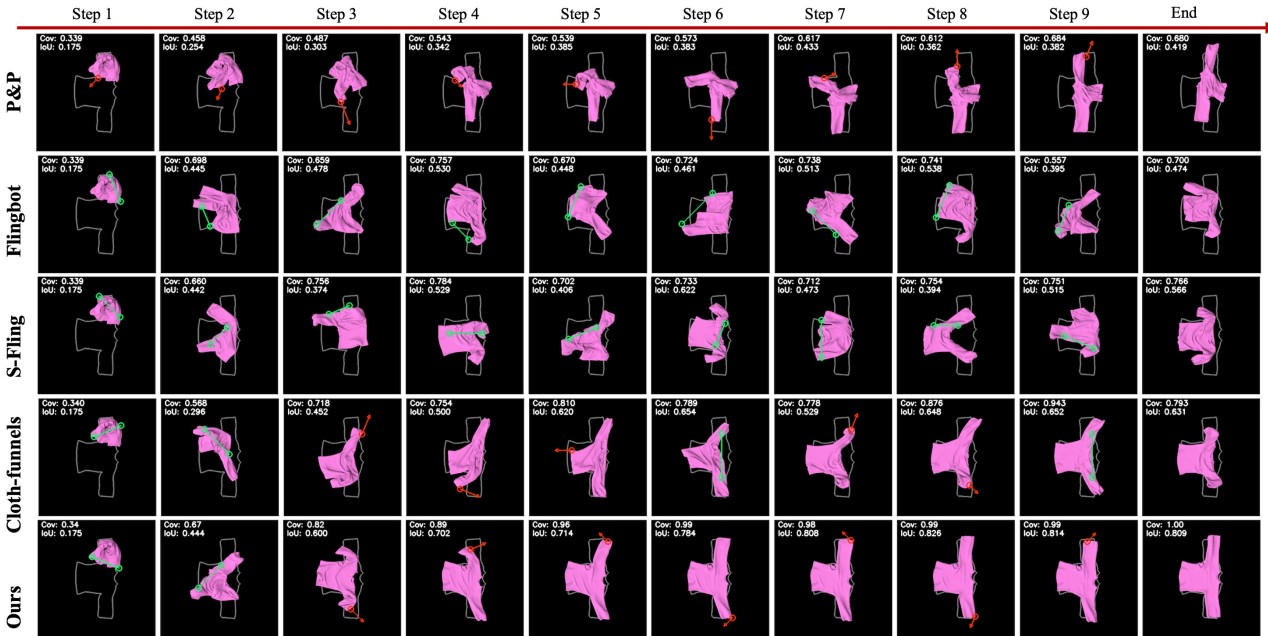

*Figure 6.* Qualitative comparison of simulation unfolding. Our method outperforms the four other methods. The red arrow denotes p&p action, while the green line indicates a fling action. The gray contour line represents the standardized garment template.

*Table 1.* Results of our method and four baselines.

| TYPE | COV(%)↑ | IoU(%)↑ | KD↓ | $R_{\text{CIK}}$↓ |
|---|---|---|---|---|
| P&P | 61.3 | 51.6 | 2.759 | 0.261 |
| S-FLING | 80.5 | 63.3 | 1.892 | 0.135 |
| FLINGBOT | 80.1 | 60.9 | 2.084 | 0.086 |
| CLOTH FUNNLES | 87.2 | 74.1 | 1.972 | 0.058 |
| OURS | **91.1** | **79.3** | **1.832** | **0.042** |

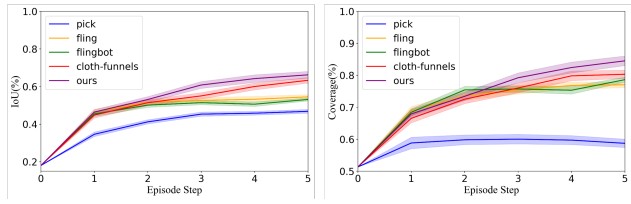

*Figure 7.* Garment Unfolding Coverage and IoU vs. Steps.

proaches. **Quasi-static Methods** include **p&p**, where the garment is sequentially lifted and placed, similar to (Lee et al., 2021). **Dynamic Methods** consist of **FlingBot**(Ha & Song, 2022), which uses dual-arm dynamic flinging with coverage as the reward function, and **S-fling**, a variant of our method using a single fling action. **SOTA Methods** include **Cloth funnels**(Canberk et al., 2023), which combines p&p and fling actions, utilizing factorized distances between cloth particles for unfolding.

### 4.3. Results and Analysis

Due to the complex shape and high manipulation difficulty of long sleeves garments, we selected them as the target for our experiments. Each policy was evaluated through 50 trials with random initial configurations of the garments.

For unfolding, Table 1 shows the quantitative results, while

Figure 6 and Figure 7 present representative qualitative results. A comparison between S-fling, Flingbot, and p&p reveals that the former two methods significantly improve coverage due to the dynamic nature of the fling actions. However, while coarse-grain high-velocity actions are effective for unfolding, they are not versatile enough for standardization. Cloth funnels outperform S-fling in coverage and IoU by facilitating finer adjustments with quasi-static actions. However, its reward function, based on the 3D distance between cloth particles, can lead to local minima during policy learning, limiting its ability to achieve optimal unfolding. In contrast, our method introduces a novel 2D plane-based factorized reward function, which achieves higher coverage and IoU with fewer manipulation steps, ensures garment standardization, and demonstrates the synergy between coarse-grain high-velocity flings and fine-grain p&p actions for optimal performance.

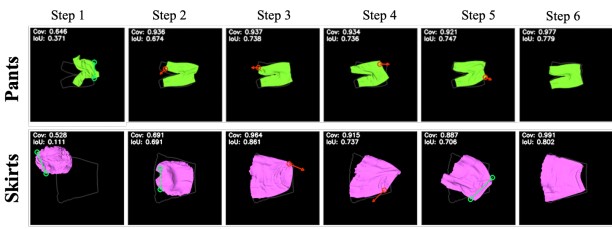

*Figure 8.* Unfolding results for different garment types.

*Table 2.* Performance metrics for Pants and Skirts.

| TYPE | COV(%)↑ | IOU(%)↑ | KD↓ |
|---|---|---|---|
| PANTS | 91.8 | 80.4 | 1.819 |
| SKIRTS | 86.7 | 79.6 | 1.825 |

We trained models on two additional garment categories—pants and skirts—using the same configuration(See Figure 8). As shown in Table 2, our method achieves around 80% IoU across all categories, demonstrating its generalizability.

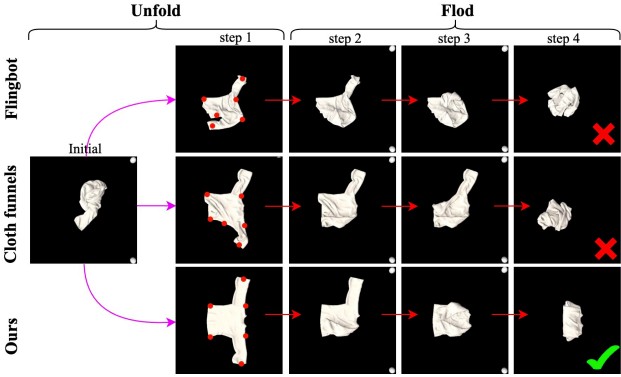

*Figure 9.* **Simulation Folding Qualitative Comparison.** We compare the unfolding results of our method with those of other existing methods using our folding heuristic.

For folding, we compare our method with existing approaches. Our method achieves superior folding performance(See Figure 9). Unlike methods focused solely on maximizing coverage, standardized unfolding aligns the garment's shape and keypoints, which simplifies the folding process and making it well-suited for garment packing in factory settings. In our experiments, we set the hyperparameters to $\alpha = 0.2$ and $\beta = 0.4$. To assess the sensitivity of our method to these parameters, we conducted a series of experiments with different values of $\alpha$ and $\beta$ (See Table 3).

*Table 3.* Effect of different hyperparameters $\alpha$ and $\beta$.

| PARAMETER | COV(%)↑ | IOU(%)↑ | KD↓ |
|---|---|---|---|
| $\alpha = 0.2, \beta = 0.2$ | 89.9 | 76.5 | 1.844 |
| $\alpha = 0.2, \beta = 0.4$ | 91.1 | 79.3 | 1.832 |
| $\alpha = 0.4, \beta = 0.2$ | 90.7 | 78.6 | 1.841 |
| $\alpha = 0.4, \beta = 0.4$ | 89.4 | 77.5 | 1.863 |
| $\alpha = 0.6, \beta = 0.2$ | 88.7 | 77.9 | 1.869 |

*Table 4.* Ablation Studies.

| METHOD | COV(%)↑ | IOU(%)↑ | KD↓ |
|---|---|---|---|
| W/O KPR | 88.2 | 76.4 | 1.851 |
| W/O AOM | 89.5 | 77.6 | 1.860 |
| W/O IR | 90.3 | 72.2 | 1.994 |
| OURS | **91.1** | **79.3** | **1.832** |

### 4.4. Ablation Studies

To evaluate the importance of different components in our method, we conduct ablation experiments by comparing our method with the following, with 50 trials per ablation using random configurations of long sleeves garments:

- **W/O AOM:** Excludes the Action Optimized Module, resulting in no optimization for the fling action.

- **W/O IR:** Excludes the IoU Reward, which helps the garment achieve better coverage, potentially reducing accuracy in spatial matching.

- **W/O KPR:** Removes the KeyPoint Reward, which directly aids in keypoint detection, potentially impairing performance in precise keypoint localization.

Table 4 shows quantitative comparisons with ablations. Clearly, each component improves our method's capability.

*Table 5.* Results of real world experiments on Garment Unfolding.

| METHOD | IOU (%)↑ | COV (%)↑ |
|---|---|---|
| P&P | 43.7 | 52.3 |
| FLINGBOT | 51.3 | 72.8 |
| CLOTH FUNNELS | 65.2 | 82.2 |
| OURS | **70.5** | **86.3** |

### 4.5. Real World Results

In our real-world experiments, we performed unfolding and folding tasks on long sleeve garments, conducting 15 trials for each group, with models trained in a simulated environment. For the unfolding task, quantitative results are presented in Table 5. Our method achieved the best performance, reaching 70.5% in IoU and 86.3% in Cov, while preserving the standardization of garment , thus validating the conclusions drawn from the simulation.

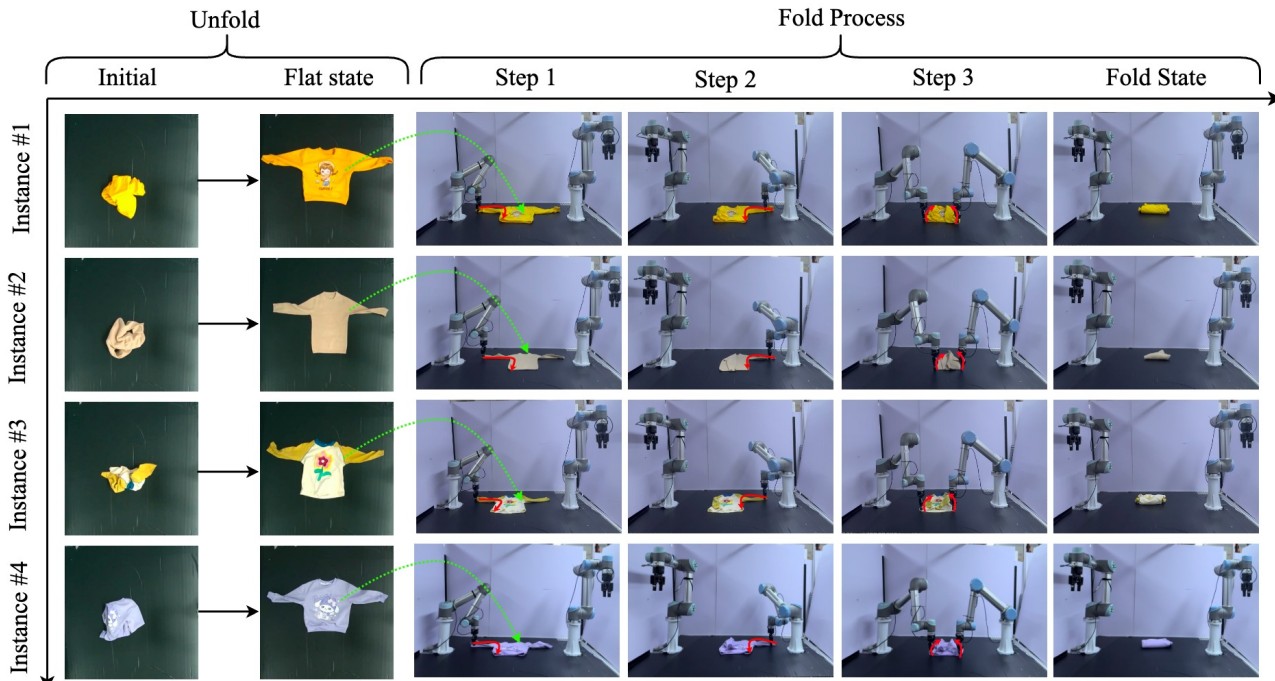

*Figure 10.* Real-world Folding.

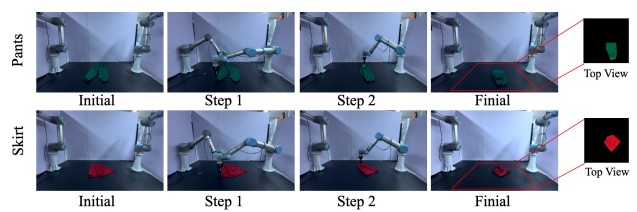

*Figure 11.* Real-world Folding in pants and skirts.

*Table 6.* Success rates across different methods.

| METHOD | SUCCESS RATE |
|---|---|
| P&P | 1/15 |
| FLINGBOT | 6/15 |
| CLOTH FUNNELS | 10/15 |
| OURS | **12/15** |

We applied our keypoint-based folding model to various unfolding methods, as shown in Table 6 (quantitative results) and Figure 10 (qualitative results). Our model achieved a success rate of 12 out of 15, demonstrating superior performance compared to existing methods. Furthermore, the results demonstrate that standardization reduces the complexity of downstream tasks.

Additionally, we conducted experiments on pants and skirts

*Table 7.* Results of real world experiments on other garment types.

| METHOD | SLEEVE | PANTS | SKIRT |
|---|---|---|---|
| FOLD | 12/15 | 14/15 | 11/15 |

(See Figure 11 and Table 7). The results demonstrate that our method generalizes to different types of garments.

## 5. Conclusion

In this paper, we presented APS-Net, a novel approach for garment manipulation that integrates both unfolding and standardization into a unified framework. By leveraging a dual-arm, multi-primitive policy, APS-Net efficiently unfolds crumpled garments and ensures precise standardization, which simplifies downstream tasks like folding. We introduced a factorized reward function, spatial action mask, and an action optimized module to enhance performance. Experimental results show that APS-Net outperforms existing methods in both simulation and real-world folding tasks, demonstrating its potential for practical garment manipulation. Our approach provides a solid foundation for further advancements in robot-assisted garment handling for a range of applications.

## Impact Statement

This paper presents a method for garment manipulation. It has potential applications in household and industrial automation, reducing manual labor. We have not identified any particular ethical issues that need to be emphasized.

## Acknowledgements

This work was supported in part by the National Natural Science Foundation of China (No. 62088101), in part by the Science and Technology Commission of Shanghai Municipality (No. 2021SHZDZX0100, 22ZR1467100), and the Fundamental Research Funds for the Central Universities (No. 22120240291).

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

## A. Factorized Reward Function Details

The Coverage Reward $R_C$ measures the proportion of newly covered garment area relative to the total garment area, promoting actions that maximize garment coverage:

$$R_C = \frac{C}{C_{\max}} \quad (16)$$

where $C$ is a function that computes the current area of the cloth based on the image pixels, and $C_{\max}$ is the maximum area of the garment.

The IoU Reward $R_I$ evaluates the overlap between the current garment configuration $m_c$ and the target configuration $m_{\text{gt}}$, ensuring proper alignment:

$$R_I = \frac{|m_c \cap m_{\text{gt}}|}{|m_c \cup m_{\text{gt}}|} \quad (17)$$

The Keypoint Reward $R_K$ is designed to evaluate the accuracy of key garment feature positioning by minimizing the distance between the predicted and target keypoints.

$$R_K = -\sum_{i=1}^{N} \|p_i - p_i^*\| \quad (18)$$

where $p_i$ represents the current position of keypoint $i$, $p_i^*$ is its target position, and $N$ is the total number of keypoints.

## B. Spatial Action Mask Details

Directly sampling actions directly on the existing spatial policy maps will lead to many invalid actions, such as empty actions in parts other than garments. The shape of the garment mask varies due to differences in rotation and scaling. To address this, we introduce the Spatial Action Mask (SAM) to filter out invalid actions, ensuring that the robot only considers feasible actions within its workspace.

The robot's arm reach is constrained by its physical limits, which are represented by masks in the simulation. The workspace is defined by a table width $w = 1.53$ m, with the left and right arm bases at positions $[0.765, 0, 0]$ and $[-0.765, 0, 0]$, respectively. The maximum reachable distance of a single arm is denoted as $r_d$. The pixel radius $r_{\text{pix}}$ is calculated by scaling $r_d$ relative to $w$, as:

$$r_{\text{pix}} = \lfloor D \cdot r_d / w \rfloor \quad (19)$$

where $D$ is the rendered image dimension. Circular masks for both arms are created, with the left arm centered at the upper half and the right arm at the lower half of the image. These masks are drawn in white, indicating reachable areas (see Figure 12 (a), (b)).

**Workspace Mask.** Both the p&p and fling actions rely on workspace masks to filter the valid action space for the robot. For the p&p action, only one arm is used at a time. The valid action space is determined by the union of the workspace masks for the left and right arms. To account for each arm's reach, the workspace masks for both arms are each shifted down by $w$ pixels, as described in formula (9) (see Figure 12(c), (d)). The final p&p workspace mask $m_w^{(p\&p)}$ is obtained by taking the union of the shifted masks (see Figure 12(e)).

For the fling action, both arms are used simultaneously. The valid action space is determined based on the position of the grabbing point $p_1$. If $p_1$ corresponds to the left arm, the left arm's workspace mask is shifted upwards by $w$ pixels, and the right arm's workspace is shifted downwards by $w$ pixels. The valid action space is the intersection of these two shifted masks (see Figure 12(j)). If $p_1$ corresponds to the right arm, the left arm workspace mask is shifted downwards by $w$ pixels, and the right arm workspace mask is shifted upwards by $w$ pixels. The valid action space is the intersection of these two shifted masks (see Figure 12(k)). The final fling workspace arm mask $m_w^f$ is obtained by taking the union of these two intersection masks (see Figure 12(l)).

The workspace mask for the action $M_{\text{w\_mask}}$ can thus be represented as:

$$M_{\text{w\_mask}} = \begin{cases} m_w^{(p\&p)} & \text{if m = p\&p} \\ m_w^f & \text{if m = fling} \end{cases} \quad (20)$$

**Garment mask.** The valid garment mask for different action primitives is computed as follows. In the *p&p* action, since the grabbing points are directly on the garment, the valid garment mask is simply the original garment mask $m_g$. For the fling action, the grabbing points for both arms are determined using formula (9). To ensure that these points lie within the garment mask, the original garment mask is shifted downward by $w$ pixels for the left arm and upward by $w$ pixels for the right arm. These shifted masks are shown in green and blue (see Figure 13(c)). The final fling garment mask $m_g^f$ is obtained by taking the intersection of the original garment mask with both the upward and downward shifted masks (see Figure 13(d)). Therefore, the garment mask for action $M_{\text{g\_mask}}$ is:

$$M_{\text{g\_mask}} = \begin{cases} m_g & \text{m = p\&p} \\ m_g^f & \text{if m = fling} \end{cases} \quad (21)$$

The valid action space is determined by the intersection of the garment mask $M_{\text{g\_mask}}$ and the workspace mask $M_{\text{w\_mask}}$:

$$M_t = M_{\text{g\_mask}} \cap M_{\text{w\_mask}} \quad (22)$$

This combined mask $M_t$ filters the action probabilities output by APS-Net. If an action $a_t$ falls within the valid region ($M_t(a_t) = 1$), it is retained; otherwise, its probability is set

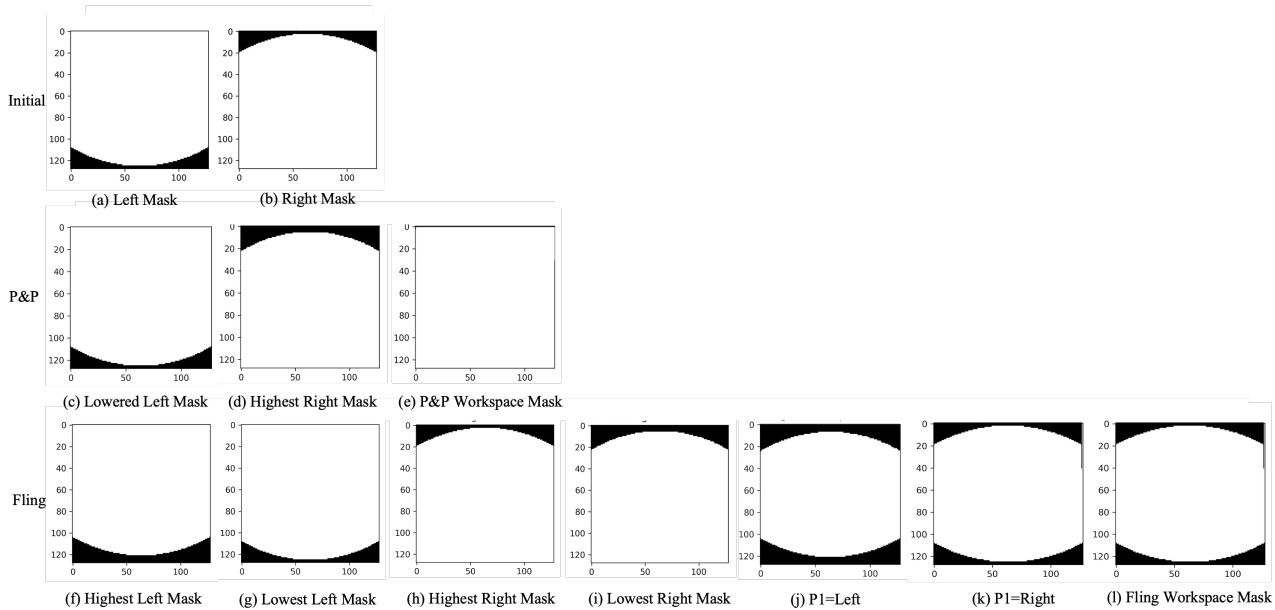

*Figure 12.* Dual arm workspace for different action primitives.

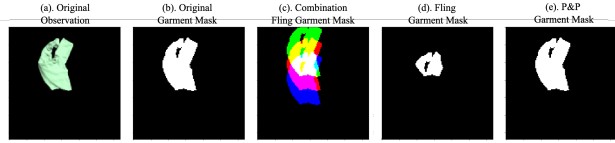

*Figure 13.* Garment for different action primitives.

to $-\infty$, excluding it from consideration. The final action probability $\pi_\theta^*(a_t|o_t)$ is:

$$\pi_\theta^*(a_t|o_t) = \begin{cases} \pi_\theta(a_t|o_t) & \text{if } M_t(a_t) = 1, \\ -\infty & \text{if } M_t(a_t) = 0. \end{cases} \quad (23)$$

## C. Experiment Setup.

### C.1. Simulation Setup

We built upon the OpenAI Gym API(Brockman et al., 2016) and PyFleX(Li et al., 2018) bindings to Nvidia FleX, integrated with SoftGym(Lin et al., 2021), to develop a reinforcement learning environment called Cloth Action Gym, which supports loading arbitrary cloth meshes, such as T-shirts, pants, and skirts, through a Python API. The robot gripper is modeled as a spherical picker that can move freely in 3D space and can be activated to attach to the nearest particle. Observations are rendered in Blender 3D, with the cloth's color uniformly sampled across specified HSV ranges.

### C.2. Real World Setup

Our experimental setup comprises two 6-DoF UR5 robot arms, each equipped with a Robotiq 2F-85 gripper, positioned 1.45 meters apart and facing each other. Two top-down RGB-D cameras—an Azure Kinect and a Realsense D455—capture the workspace. The Grounded-SAM model (Liu et al., 2023; Kirillov et al., 2023) is used to segment the garment from the RGB image with the prompt "garment." To minimize the domain gap between simulation and real-world images, we apply an Affine Transform to crop and correct the camera's perspective, focusing on the central region of the garment. The resulting mask is then multiplied by the RGBD image to generate the input for the model. (See Figure 14) This approach allows the simulation-trained model to be effectively adapted for real-world use without requiring additional training.

## D. Data Collection

In the simulation environment, we collected two datasets, one for training garment unfolding and another for training garment keypoint detection. Each task is characterized by specific parameters such as the cloth mesh, mass, stiffness, and initial configuration. The cloth meshes were sampled from a subset of shirts in the test split of the CLOTH3D dataset(Bertiche et al., 2020), resized to ensure they fall within the robot's operational range. The cloth mass was selected from a range of $[0.2\,\text{kg}, 2.0\,\text{kg}]$, and internal stiffness was set between $[0.85\,\text{kg/s}^2, 0.95\,\text{kg/s}^2]$. To initial-

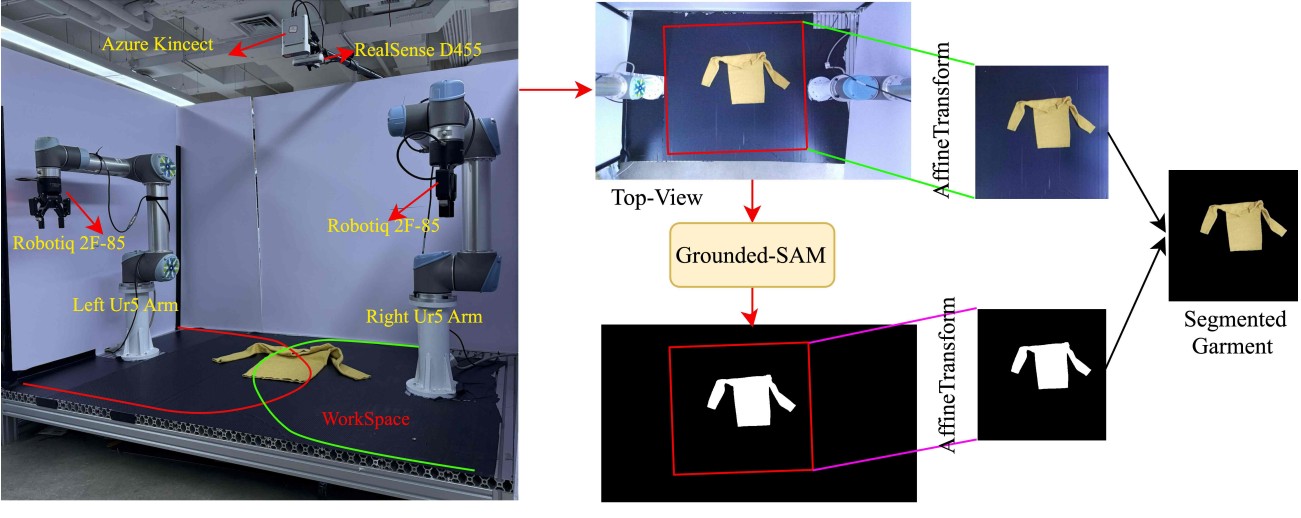

*Figure 14.* Real-world Setup.

ize the cloth states, we randomly sample from a subset of CLOTH3D, which provides a curated collection of garment meshes.

For the garment unfolding dataset, the initial configuration was created by randomly rotating the garment, selecting a random point on the garment, and dropping it from a height between $[0.6\,\text{m}, 1.6\,\text{m}]$. The garment was then translated by a random distance in the range of $[0\,\text{m}, 0.25\,\text{m}]$, resulting in a highly crumpled configuration. Each garment category includes 2000 training tasks and 50 testing tasks using unseen garment meshes.

For the garment keypoint detection dataset, we filtered images from the replay buffers generated during the unfold task, selecting images where the coverage exceeded $0.8$. This process resulted in a dataset of 10000 images for training and 500 for testing. Real-world data collection involved 1100 images, which were labeled using LabelMe.

## E. Training Details

For the unfolding network, the initial observation $o_t$ of size $(H, W) = (128, 128)$ undergoes 16 rotations (covering $360°$) and 5 scales $\{0.75, 1.0, 1.5, 2.0, 2.5\}$, yielding $m = 80$ stacked transformed observations. Exploration follows a decaying $\epsilon$-greedy strategy (initial $\epsilon = 1$) with half-lives of 5000 steps for selecting action primitives (fling vs. p&p) . Optimization is conducted using the Adam optimizer with a learning rate of $1.0 \times 10^{-3}$ over 100,000 steps. The value network is supervised using delta-reward values, defined as the difference in weighted reward ($\Delta R_{CIK}$) before and

after each action:

$$\Delta R_{CIK} = R_{CIK,\text{after}} - R_{CIK,\text{before}} \quad (24)$$

The value network minimizes the Mean Squared Error (MSE) loss between the ground truth delta-reward $\Delta R_{CIK}^{\text{gt}}$ and the predicted delta-reward $\Delta R_{CIK}$:

$$L_{\text{MSE}} = \frac{1}{N} \sum_{i=1}^{N} (\Delta R_{CIK}^{\text{gt}} - \Delta R_{CIK})^2 \quad (25)$$

For the key-point detection model, training is conducted for 200 epochs with a batch size of 32, a learning rate of $1.0 \times 10^{-4}$, and includes rotation and scaling augmentations. The training loss $L(y, p)$ is defined as the mean point-wise cross-entropy loss for each keypoint $n$ in $N$:

$$L(y, p) = -\frac{1}{N} \sum_{n=1}^{N} \big(y_n \log(p_n) + (1 - y_n) \log(1 - p_n)\big) \quad (26)$$

where $y_n$ is the true label and $p_n$ is the predicted probability for each keypoint $n$. All experiments are conducted on an NVIDIA RTX 4090 GPU with an Intel i9-13900K CPU (5.80 GHz), supported by 64 GB RAM on Ubuntu 18.04 LTS.

## F. Experimental Results

Figure 15 shows how our method transfers the garment from an arbitrary configuration to a standardized one. Figure 16 presents the qualitative results of garment unfolding in the

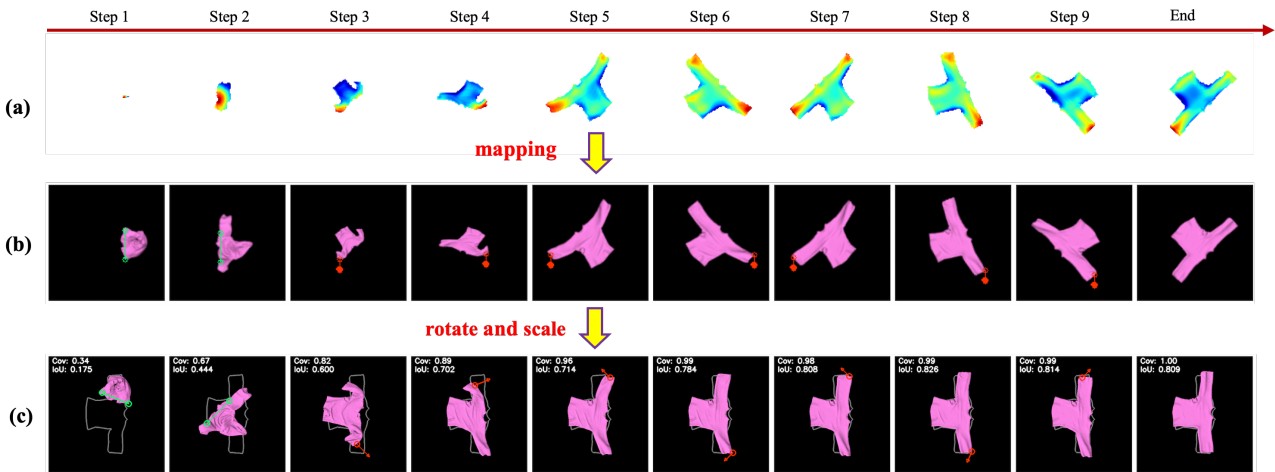

*Figure 15.* Our method transfers the garment from an arbitrary configuration to a standardized configuration. The first row shows the spatial action map output by the APS-Net network at each step. The second row maps the highest pixels of the spatial action map to the transformed image at each step, producing the corresponding actions. The third row applies rotation and scaling at each step to transform these actions back to the original input image.

real world, comparing our method with other approaches. In Figure 17, we compare the unfolding results of our method with those of Cloth Funnels using our folding heuristic. Finally, Figure 18 compares optimized and unoptimized shoulder keypoint alignment for garment flattening.

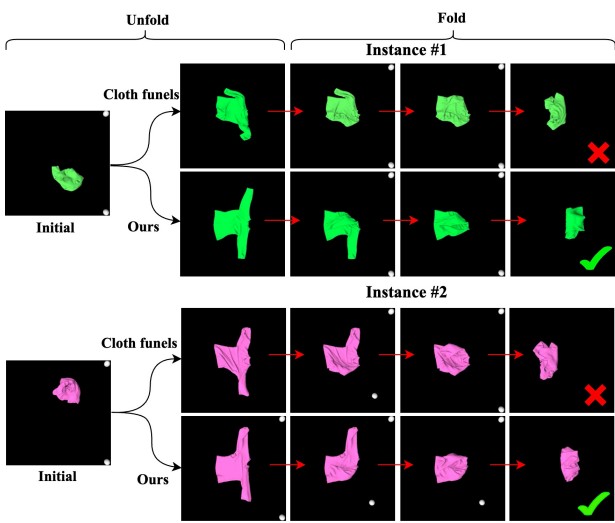

*Figure 17.* We compare the unfolding results of our method with those of Cloth Funnels using our folding heuristic.

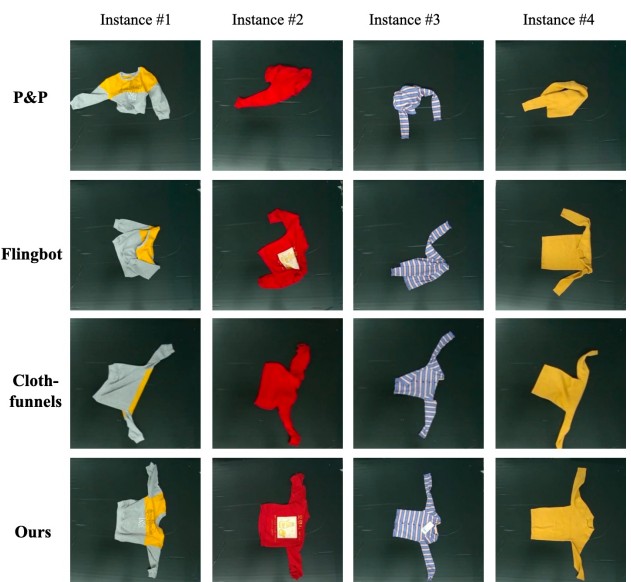

*Figure 16.* **Qualitative results of garment unfolding in real world.** Our method outperforms three other methods on long sleeve garments with four different colors and textures.

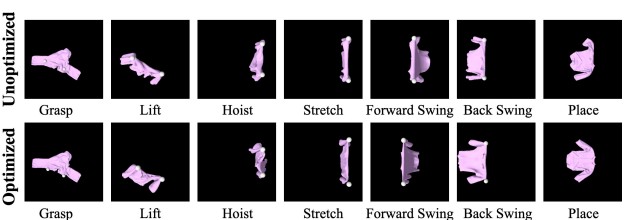

*Figure 18.* Comparison of Optimized vs. Unoptimized Shoulder KeyPoint Alignment for Garment Flattening.

