# OpenReview forum: "Learning Efficient Robotic Garment Manipulation with Standardization"
_ICML.cc/2025/Conference — ICML 2025 poster_

### Official Review · Reviewer_BU25 · 2025-03-10

**Overall Recommendation:** 2

**Summary:**

The authors present APS-Net, a unified framework for garment manipulation that integrates both unfolding and standardization. APS-Net employs a dual-arm, multi-primitive policy to unfold crumpled garments and ensure standardization, which facilitates downstream tasks like folding. Experimental results show that APS-Net outperforms baselines in both simulation and real-world tasks.

**Claims And Evidence:**

Yes

**Essential References Not Discussed:**

No.

**Experimental Designs Or Analyses:**

See Other Strengths And Weaknesses.

**Methods And Evaluation Criteria:**

Yes

**Other Comments Or Suggestions:**

In Line 262, a space is missing after the period.

**Other Strengths And Weaknesses:**

Strengths

1. The authors conduct real-world experiments and provide detailed analysis for experiments.

2. APS-Net introduces a dual-arm, multi-primitive policy and integrates garment unfolding and standardization into a unified framework, offering notable improvement over prior work such as P&P and FlingBot.


Weaknesses

1. The proposed approach is straight-forward and complicated. The framework is designed around three metrics (coverage, IoU, keypoint distance), with a corresponding three-encoder, six-decoder architecture (each encoder is paired with both fling and P&P decoders). The contribution and novelty in the pipeline architecture may be limited.

2. While the Introduction is well-written and easy to follow, the Methods section could be improved for clarity, as some details are confusing (see Questions).

3. The same metrics (coverage, IoU, keypoint distance) used in the experiments are also employed during the training of APS-Net. It is unclear whether these metrics are also used in the training of other baseline methods. If not, the comparison might be unfair.

**Questions For Authors:**

1. What does $k$ represent? The paper does not define this variable.
2. Figure 15: In Step1 and Step2, why do the shapes of the spatial action maps in the first row not match the shapes in the second row?
3. The text and equations suggest that value maps are the aggregations of scores for each $<x, y, \theta, w>$, implying a high-dimensional representation (dimensionality > 3). However, the visualized value maps in Figure 2 appear to be three-dimensional (xyz space), which is confusing. Could the authors provide some clarifications? Additionally, should θ be discrete? This is not explicitly mentioned how $theta$ is handled.
4. Section3.2 explains how the grasp points are generated, but does not explain how place points are determined.
5. The concept of 'midpoint (x, y) between two grasp points' is used for pick&place action (in Euqation9), however, it seems only one arm is used for pick&place action, so it is unclear what the midpoint represents in this context.

**Relation To Broader Scientific Literature:**

Compared to previous works, APS-Net integrates both garment unfolding and standardization into a unified framework.

**Theoretical Claims:**

Yes

---

> ### Author Rebuttal · Authors · 2025-04-01
>
> We sincerely thank you for your thoughtful feedback, insightful questions, and recognition of our work’s strengths. Below we address the comments point-by-point:
>
> ## Weaknesses
>
> ### 1. Framework Architecture Complexity
>
> We understand the concern about APS-Net's complexity. However, it is specifically designed for the factorized reward function, with each component essential for guiding fling and p&p actions in garment standardization. Below, we explain its necessity and novelty.
>
> The three encoders (coverage, IoU, and keypoint distance) directly target the core challenges in garment manipulation:
> - Coverage Encoder: Guides the fling action to quickly unfold crumpled garments.
> - IoU Encoder: Guides p&p action fine-grained adjustments near-flattened garments for consistent shape and orientation.
> - Keypoint Encoder: Ensures visibility of key garment features (e.g., collar, sleeve, hem)
>
> Removing any encoder causes a performance drop, as shown in ablation studies (Table 4). Thus, this design is not arbitrary, but a novel architecture tailored to garment unfolding.
>
> To further validate our design, we conducted experiments with an unfactorized network structure (a single-encoder variant with two decoders (fling and p&p) and a weighted sum of the same metrics), resulting in poorer performance (see new Table 4.1).
>
> Table 4.1 Unfactorized network structure
> | Method       | COV (%) | IOU (%) | KD     |
> |--------------|---------|---------|--------|
> | Unfactorized | 87.5    | 73.7    | 1.995 |
> |||
>
> ### 3. Clarifying Metric Usage
>
> We used the same reward function for FlingBot and P&P baseline as ours, with results shown in Table 1 under "S-Fling" and "P&P." We also compare this with Flingbot's original method, which used only coverage as the reward. These comparisons ensure a fair evaluation.
>
> ## Questions
>
> ### 1. Undefined Variable (k)
>
> The variable k was a typo; the correct variable is m (m ∈ {fling, p&p}). This will be corrected in the revision.
>
> ### 2.Spatial Action Map Shapes in Figure 15
>
> The shape differences in spatial action maps arise from applying a spatial action mask (SAM), detailed in Section 3.4 and Appendix B.
>
> For dual-arm fling, SAM filters infeasible actions, such as regions with potential arm collisions or where only one arm can grasp the garment. For example, Figures 13(a) and 13(b) show the original garment and corresponding mask, while 13(d) shows the mask applied, excluding infeasible regions. Therefore, the valid mask does not match the garment's original shape.
>
> For single-arm p&p, collision avoidance is unnecessary, so the mask is aligned to the shape of the garment (see Figure 13(e)).
>
> ### 3.Dimension Gap and Meaning of θ
>
> The value maps are 3D, with the third dimension representing the number of transformations $T_n$ (scaling and rotation). The correct formulation for equation (8) is:
> $$
> ⟨x, y, i⟩ ← APSNet(T_n) = argmax(V_{f_{\max}}^{\max})
> $$
> Once i is determined, the values of w and θ are retrieved from the i-th entry of $T_n$:
> $$
> w, θ = T_n[i]
> $$
>
> Based on the above reasoning, we obtain ⟨x, y, θ, w⟩.
>
> As for θ, it is indeed discrete. In Appendix E, we mention that θ undergoes 16 rotations, covering 360°.
>
> ### 4.Determination of Place Points
>
> Based on the definition (see Figure 1), for single-arm p&p, both the grasp and place points need to be predicted. As defined in Equation (9) (when m = p&p), the grasp point is (x, y), and the place point is (x − w, y).
>
> For dual-arm fling, only two grasp points are predicted, while place points follow a predefined swinging trajectory (Equation 4.1). As defined in Equation (9) (when m = fling), the grasp points are (x + w, y) and (x − w, y). During execution, both arms follow the trajectory (lift, forward/backward motion, place), adjusting acceleration and velocity at each stage to complete the fling motion.
>
> $$
> fling = [(0,0,h_l) \to (0,f,h_l) \to (0,f-b,h_l) \to (0,f,h_p)]  \quad \text{4.1}
> $$
>
> where $h_l$ is the lift height, $f_m$ is the forward swing distance, $b_m$ is the backward swing range, and $h_p$ is the placement height.
>
> ### 5.Clarifying Midpoint in Equation (9)
>
> For dual-arm fling, directly predicting both grasp points is challenging due to collision avoidance. Instead, we reframe the problem by predicting the midpoint (x, y), angle θ (rotation), and grasp width w, with collision avoidance achieved by adjusting w.
>
> For single-arm p&p, collision avoidance is not required. However, to maintain consistency with dual-arm fling and provide a unified output from the same network, we define (x, y) as the pick point, while θ and w represent the rotation and length between the pick and place points.
>
> Therefore, we generalize the two cases as solving for ⟨x, y, θ, w⟩, and determine the operation points for two action primitives through Equation (9).
>
> ## Suggestions
>
> ### Space missing
>
> We will correct this in the revised manuscript and will be more attentive to formatting details in the future.

---

> > ### Comment · Reviewer_BU25 · 2025-04-02
> >
> > Thank you for your detailed response. The clarifications and explanations provided have addressed most of my concerns.
> >
> > However, based on the results from new Table 4.1, it appears that the architecture of the three-encoder, six-decoder setup plays a more important role than the weighted sum of the metrics (i.e., the reward function). Additionally, the results of this new ablation study appear to be indistinguishable from the baseline CLOTH FUNNELS (Table 1). This may suggest that the performance of the proposed method is more attributed to technical refinements rather than an innovative contribution. Could the authors provide further clarification on this point?

---

> > > ### Author Response · Authors · 2025-04-04
> > >
> > > Thank you very much for taking your valuable time to read my response and acknowledging most of my previous clarifications. We sincerely apologize for previously not addressing your concern sufficiently clearly. Below, we provide a more detailed clarification regarding your specific question.
> > >
> > > We agree with your insightful observation based on Table 4.1 that the three-encoder, six-decoder architecture contributes to overall performance. However, we would like to emphasize that the proposed weighted reward function plays the central role in driving the improvements. To demonstrate this more clearly, we have conducted additional ablation experiments (see new Table 4.2), where the model is trained using only individual reward metrics, rather than the proposed weighted combination. These experiments showed a substantial drop in performance compared to the results obtained using the weighted sum approach. Therefore, although the architecture contributes to the overall performance, the weighted reward function remains essential and has a dominant effect on the model’s behavior. Moreover, the architecture was specifically designed to support the weighted reward, serving as a tailored solution for learning from complex visual signals. Together, the architecture and reward function constitute complementary innovations at the core of our method.
> > >
> > >
> > > Table 4.1 Unfactorized network structure
> > >
> > > | Method       | COV (%) | IOU (%) | KD     |
> > > |--------------|---------|---------|--------|
> > > | Unfactorized | 87.5    | 73.7    | 1.995  |
> > > |||||
> > >
> > >
> > >
> > > Table 4.2 Individual Reward Metrics.
> > > | Method | COV (%) | IOU (%) | KD    |
> > > |--------|---------|---------|--------|
> > > | Cov    | 92.6    | 53.4    | 2.698  |
> > > | IOU    | 79.9    | 66.2    | 2.583  |
> > > | KEYPOINT    | 68.9    | 55.1    | 2.621  |
> > > |||||
> > >
> > > Regarding your concern about the apparent similarity between the performance of our ablation study (Table 4.1) and the CLOTH FUNNELS baseline, we would like to clarify the following issue. In Table 4.1, it used a much simpler architecture—only one encoder and two decoders—compared to CLOTH FUNNELS yet achieved comparable performance. This result demonstrates that, even when using a simplified architecture, the weighted reward method can maintain performance at the baseline level. More importantly, when employing our full architecture (three encoders and six decoders) with the proposed weighted reward performance significantly surpasses CLOTH FUNNELS (see Table 1). This level of improvement is unlikely to be achieved by minor technical optimizations or training tricks alone.
> > >
> > > Furthermore, our method leverages a weighted combination of evaluation metrics, enabling it to compute rewards directly from visual observations (RGB-D data). In contrast, prior approaches such as Lin et al. [1], CLOTH FUNNELS [2], and Deng et al. [3] rely on particle-based cloth representations to compute rewards or losses for cloth manipulation tasks. While such particle data is readily available in simulated environments like SoftGym, it is difficult to acquire in real-world. By relying solely on visual inputs, our method avoids the sim-to-real gap introduced by inaccessible simulation-specific data, resulting in better generalization and improved robustness in real-world cloth manipulation tasks. Therefore, our approach should not be regarded as a minor technical refinement, but rather as a principled and practical innovation that overcomes fundamental limitations of prior work.
> > >
> > > Importantly, our method naturally reveals garment key points during the flattening process. This significantly simplifies the challenge posed by infinite degrees of freedom. Furthermore, identifying these garment key points is crucially beneficial to downstream tasks, such as garment dressing and hanging, thereby clearly demonstrating the broader applicability and scalability of our method compared to existing approaches.
> > >
> > > In summary, our method is not a minor technical adjustment, but a response to fundamental challenges in cloth manipulation, integrating reward design and architecture to offer a more robust and generalizable solution.
> > >
> > > We sincerely thank the reviewer once again for the thoughtful and constructive feedback. Your comments have helped us to more clearly articulate the contributions and motivations of our work. We greatly appreciate your time and effort in reviewing our paper. thank you!
> > >
> > > ### Reference
> > >
> > > [1] Lin X, Wang Y, Huang Z, et al. Learning visible connectivity dynamics for cloth smoothing. CoRL 2022.
> > > [2] Canberk A, Chi C, Ha H, et al. Cloth funnels: Canonicalized-alignment for multi-purpose garment manipulation. ICRA 2023.
> > > [3] Deng Y, Mo K, Xia C, et al. Learning language-conditioned deformable object manipulation with graph dynamics. ICRA 2024.

---

### Official Review · Reviewer_HhgQ · 2025-03-13

**Overall Recommendation:** 3

**Summary:**

In this paper, the authors present an RL framework for grament manipulation, which consists of two stages: standardization and folding. For the standardization stage, a two-primitive polcy of fling and pick-and-place is trained using a factorized reward function, which includes garment converage, keypoint distance and IOU. After standardization, folding is performed using a keypoint detection-based method. Two real-world datasets are collected to improve the real-world performance.

The authors compare their method with four baseline methods in simulation, and their method achieves SOTA in three metrics. The authors further conduct ablation studies to verify the effectiveness of each module. And they condut unfolding and folding experiments in the real world. Their method outperforms other methods in terms of success rate.

**Claims And Evidence:**

Yes, the effectivness of their method and each module are verified through experiments and ablation studies.

**Essential References Not Discussed:**

NA

**Experimental Designs Or Analyses:**

The experimental designs are good and extensive. And the analysises verify the effectivenss of their method.

**Methods And Evaluation Criteria:**

The main novelty of this method is the learning-based primitive selection and the factorized reward function. It can be considered as a technical improvement over existing pipeline, while there is no innovative contribution. And the performances in simulation and realworld show the effectiveness of their method.

The experiments are extensive and the criterias are widely-used metrics to evaluate the performances.

**Other Comments Or Suggestions:**

The layout of the figures could benefit from improvement. Currently, it is disorganized and makes it difficult to follow the logical flow.

**Other Strengths And Weaknesses:**

The key difference between the proposed method and existing methods needs further discussion and explaination. For example, it is claimed in the related work that ``their sim-to-real gap limits real-world applicability", but it is unclear how the proposed method addreeses the domain gap in the standarization stage.

**Questions For Authors:**

What is the defination of R_c, R_I and R_K in Eq(10)?

**Relation To Broader Scientific Literature:**

NA

**Theoretical Claims:**

NO theoretical claims

---

> ### Author Rebuttal · Authors · 2025-04-01
>
> We truly appreciate the time and effort you invested in reviewing our paper. Thank you for recognizing the effectiveness of our two-stage RL framework for garment manipulation, particularly the learning-based primitive selection and factorized reward design. Below, we respond to your comments point-by-point:
>
> ### 1. Sim-to-real gap
>
> Our method minimizes this gap by incorporating several strategies to enhance simulation realism.
> Firstly, we use RGBD images rather than RGB alone, as the combination of color and depth provides a more accurate representation of the garment's position and shape. Depth data enhances spatial alignment, reducing the sim-to-real gap and improving real-world transferability.
>
> Secondly, in simulation, we use SoftGym to model cloth dynamics, with Blender 3D rendering realistic cloth colors based on HSV values sampled from real-world materials. The cloth's mass (ranging from 0.2 kg to 2.0 kg) and internal stiffness (0.85 kg/s² to 0.95 kg/s²) are varied to reflect real-world cloth properties.
>
> Thirdly, we use procedurally generated normal maps to simulate wrinkles and surface details, further enhancing simulation realism.
>
> These strategies—RGBD images, realistic cloth properties, detailed surface textures—help minimize the sim-to-real gap, ensuring effective transfer to real-world garment manipulation.
>
> ### 2. Layout of the Figures
>
> Thank you for your feedback on the figure organization. We recognize the importance of a clear and structured presentation and will revise the layout in updated version to improve the flow and organization of the figures.
>
> ### 3. Definition of  $R_c$, $R_I$, and $R_K$ in Equation (10):
>
> The terms $R_c$, $R_I$, and $R_K$ in Eq. (10) represent the different components of the factorized reward function, and their definitions can be found in Appendix A. We will update the revised version to clearly show where these definitions are located.
>
> ### 4. Clarification of the innovative
>
> We sincerely appreciate the reviewer's thoughtful feedback and recognition of our method's effectiveness. Below, we clarify the innovative aspects of our approach.
>
> Existing garment unfolding methods rely primarily on learning-based static pick-and-place strategies [1][2], requiring numerous interactions, or dual-arm approaches [3][4] maximizing coverage without standardizing garment orientation and shape—key for downstream tasks such as folding and packing. Motivated by extensive literature review and practical needs, our goal is not only to rapidly unfold garments, but to standardize their final pose.
>
> To achieve this, our proposed dual-arm, multi-primitive policy combines dynamic fling actions for efficient unfolding with precise pick-and-place operations for standardization. To effectively train this policy, we introduce a novel factorized reward function (Cov, KD, IoU metrics) and a specialized multi-encoder-decoder architecture that intelligently selects primitives based on garment state.
>
> Moreover, we incorporate a Spatial Action Mask (Section 3.4, Appendix B) to filter infeasible actions and an Action Optimization Module, enhancing fling point selection effectiveness, validated through extensive ablation studies (see Table 4).
>
> By integrating these innovations, our approach effectively generalizes across various garment types (pants, skirts, long sleeves) in Table 2. Further downstream folding experiments confirm that our standardized poses significantly improve performance (see Table 6). Finally, extensive real-world experiments on a dual-UR5 robot validate the robustness and practical applicability of our approach.
>
> ### References
>
> [1] Lin X, Wang Y, Huang Z, et al. Learning visible connectivity dynamics for cloth smoothing. CoRL 2022.
> [2] Wu R, Ning C, Dong H. Learning foresightful dense visual affordance for deformable object manipulation. ICCV 2023.
> [3] Ha H, Song S. Flingbot: The unreasonable effectiveness of dynamic manipulation for cloth unfolding. CoRL 2022.
> [4] He C, Meng L, Sun Z, et al. Fabricfolding: Learning efficient fabric folding without expert demonstrations. Robotica 2024.

---

> > ### Comment · Reviewer_HhgQ · 2025-04-03
> >
> > Thank the authors for the detailed responses. My concerns have been well addressed.

---

### Official Review · Reviewer_U7iv · 2025-03-14

**Overall Recommendation:** 3

**Summary:**

This paper introduce a novel robotic garment manipulation system with standardization, which has better performance than the previous framework in this challenging robotics task.

**Claims And Evidence:**

1. The standarization for the garment manipulation task can enhance performance.

2. However, the standarization will restrict the generalizability. Also, it's hard to set up various standarization for a lot of different kinds of clothes in the real world.

3. Detecting keypoints and selecting corresponding predefined skills can be helpful for resolving this question. However, it still restrict the generalizability.

**Essential References Not Discussed:**

No

**Experimental Designs Or Analyses:**

Conducting comprehensive experiments in both simulation and real world, for both folding and unfolding.

It's not clear how to evaluate the folding is success or not. It might be subjective？

**Methods And Evaluation Criteria:**

1. Using predefined unfolding region and manipulation skills as the standardization

2. Using proposed APS-Net to detect the fold or unfold and detect keypoints.

3. Using IoU, better coverage, and keypoint distance as input to select the correct keypoints and predefined actions.

This framework uses a lot of predefined information, which are target bounding edges and skills. It will reduce the generalizablity, and it's hard to define everything for different kinds of clothes.

The folding strategy seens very simple. It's not convincing that only initial keypoint is enough for robot to accomplish dynamic motions, where the motion of the deformable object has various uncertainty. More analysis to show that only keypoint-based strategy is enough is helpful.

**Other Comments Or Suggestions:**

The figure is too large and contains too many details... The words are too small, and it's almost impossible to see anything without zooming in on the page

**Other Strengths And Weaknesses:**

This framework uses a lot of predefined information, which are target bounding edges and skills. It will reduce the generalizablity, and it's hard to define everything for different kinds of clothes.

**Questions For Authors:**

No

**Relation To Broader Scientific Literature:**

No

**Theoretical Claims:**

Not a theoretical paper

---

> ### Author Rebuttal · Authors · 2025-03-31
>
> We sincerely thank the reviewers for their insightful feedback and for recognizing the novelty of our standardized robotic garment manipulation system. Below, we address the key concerns raised, particularly regarding generalizability, the simplicity of the folding strategy, and evaluation metrics.
>
> ### 1. Concern About Predefined Information and Generalizability
>
> Regarding target bounding edges：
> We agree that target bounding edges are used in our method, but their role is to act as an alignment metric for garment standardization. Once the model is trained, target bounding edges are no longer needed.
>
> Additionally, garments can be simply categorized into common types (e.g., skirts, pants, long sleeves), which share similar features within the same category. In simulation, garments are modeled as grids of particles, and the ground truth positions are accessible, allowing for easy computation of bounding edges. Thus, manual annotation of boundaries is not needed for different garment types. Our model, trained in simulation, generalizes across these categories, as demonstrated in our experiments on pants and skirts (see Figure 8).
>
> Regarding predefined skills:
> Selecting the appropriate action primitive is crucial in robotic manipulation, with pick-and-place being commonly used for rigid objects. For clothes, we designed two primitive actions (fling and pick-and-place) that enable rapid flattening and standardization. We tested these skills on various garment types (e.g., skirts, pants, long sleeves) without any additional parameter adjustments, and the model adapted well — demonstrating that these skills do not impact generalization.
>
> ### 2. Keypoint-Based Strategy and Generalizability
>
> Our keypoint-based reward serves as an optional metric to enhance garment feature visibility (e.g., collar/sleeve alignment) during standardized unfolding, but is not fundamental to the core method. The keypoints are obtained directly from simulation without requiring trained detectors, preserving generalizability across garment categories during training.
>
> The core contribution of our work is standardized unfolding, which benefits downstream tasks like folding, ironing, and packing. To evaluate its advantages, we used a keypoint-based folding task. However, this approach may face limitations in terms of generalizability to unseen garment categories due to structural variations in garments. Our future research will focus on adapting folding models to generalize across categories.
>
> Furthermore, even without the keypoint-based reward metric, our method performs well (see ablation study, Table 4), with generalization limits applying only to folding validation, not to unfolding.
>
> ### 3. Simplicity of the Folding Strategy
>
> As explained in Question 2, folding is not the main focus of our research; it is a validation task. To this end, we adopted a simple keypoint-based folding strategy, which is sufficient to showcase the advantages of our standardized unfolding. We believe this choice strikes a balance between simplicity and functionality for the purpose of validation.
>
> ### 4. Evaluation of Folding Success
>
> In simulation:
> The cloth is modeled as a grid of particles, and we can access the ground truth positions of these particles. We use the mean particle distance between the cloth states achieved and the desired target state. If the average particle distance error is less than 0.03, we consider the folding to be successful.
>
> In the real world:
> We evaluate performance quantitatively using the Mean Intersection over Union (MIoU) between the cloth masks achieved and the human demonstrator. If the MIoU exceeds 0.8, we consider the folding to be successful. We will clarify these evaluation metrics in the revised manuscript.
>
> ### 5. Figure Quality and Presentation
>
> We appreciate the feedback on the figure. We have simplified Figures 2 and 3 and enlarged the text for better clarity.
> The details of the changes are in the following link:  https://github.com/hellohaia/img/blob/main/1.pdf

---

> > ### Comment · Reviewer_U7iv · 2025-04-03
> >
> > Thanks for your response and clarifications. All of my questions have been addressed

---

### Official Review · Reviewer_fdEB · 2025-03-17

**Overall Recommendation:** 4

**Summary:**

This paper introduces APS-Net, a novel framework for robotic garment manipulation that seeks to both unfold garments and align them into standardized orientations—essentially “standardizing” them as part of the unfolding process. Unlike many existing solutions that focus on either single-arm quasi-static approaches or dynamic actions that only maximize coverage, APS-Net combines dynamic dual-arm fling actions for fast unfolding with more precise pick-and-place (p&p) actions for alignment. The authors propose a factorized reward function incorporating garment coverage, intersection-over-union (IoU), and keypoint positioning, guiding the system to flatten garments while preserving meaningful geometry for downstream tasks like folding

**Claims And Evidence:**

This paper claims four main things: (1) that combining fling and pick-and-place yields more efficient and accurate garment flattening, (2) that a novel factorized reward function leads to superior performance, and (3) that standardization (i.e., aligning shape/orientation) meaningfully benefits downstream tasks such as folding. Across their experiments, these claims appear largely substantiated. First, their method clearly improves coverage and IoU metrics relative to baselines like single-arm pick-and-place or exclusively fling-based approaches. The factorized reward containing coverage, IoU, and keypoint distance likewise shows consistent gains in shaping the final garment state. Ablation studies confirm that omitting parts of the reward (e.g., excluding IoU or ignoring keypoint alignment) harms performance.

**Essential References Not Discussed:**

All essential related works are discussed to the best of my knowledge.

**Experimental Designs Or Analyses:**

The experiments are structured around evaluating coverage, IoU, and keypoint distance after each rollout, alongside real robot demonstrations. The environment in simulation uses the SoftGym framework with PyFleX, which is a well-established simulator for deformable objects. The tasks are repeated with multiple random initial configurations, and baseline comparisons are made with relevant prior works (including single-arm pick-and-place, a standard fling-based method, and a state-of-the-art cloth manipulation baseline).

**Methods And Evaluation Criteria:**

The evaluation criteria include coverage, IoU, and keypoint distance; these are standard evaluation metrics in the garment manipulation literature, so they make sense.

**Other Comments Or Suggestions:**

N/A.

**Other Strengths And Weaknesses:**

Weaknesses:
1. While real-world tests are present, the paper might have benefited from more quantitative comparisons across different cloth weights or textures to fully test fling reliability.
2. The method’s reliance on overhead segmentation and keypoint detection might fail if the garment color is too close to the table or if extreme wrinkles obscure important garment keypoints. I'd like to see the method being stress tested under more extreme visual conditions.
3. Given that recent works have demonstrated competent garment folding capabilities from pure end-to-end approaches (e.g., pi-0), I think some discussions on these approaches are warranted.

**Questions For Authors:**

See my comments above.

**Relation To Broader Scientific Literature:**

This works extends a rich body of literature on robot garment folding. The system presented in this work is more "complete" and effective than prior well-known works in the literature, such as flingbot.

**Theoretical Claims:**

This is not a theory paper.

---

> ### Author Rebuttal · Authors · 2025-04-01
>
> We sincerely appreciate the time and effort you have dedicated to evaluating our work, and we are grateful for your recognition of our key contribution—the integration of dynamic flinging with precision pick&place for garment standardization, and the factorized reward function. Below, we provide a point-by-point response to the comments, incorporating additional experiments and analyses.
>
> ## Weaknesses
>
> ### 1. Fling Reliability Across Cloth Weights and Texture
>
> In simulation, we tested cloth masses ranging from 0.2 kg to 2.0 kg, with internal stiffness values between 0.85 kg/s² and 0.95 kg/s². Procedurally generated normal maps were used to simulate wrinkles and surface details. In real world, we experimented with garments of various weights, from lightweight long-sleeve shirts (e.g., Instance 2 in Figure 16) to heavier sweaters (e.g., Instance 4 in Figure 16). In terms of texture, we evaluated garments with complex patterns (e.g., Instances 1 and 3 in Figure 16). The results demonstrated that the fling motion operated reliably across all conditions. Therefore, variations in cloth weight and texture do not affect fling reliability.
>
> ### 2. Stress Testing Method under Extreme Visual Conditions
>
> We have tested our method with two garments whose colors are similar to the table surface, both in crumpled and smooth states, to evaluate the performance under extreme conditions. We tested each method 10 times on garments in both crumpled and smooth states to evaluate performance under extreme conditions. The results demonstrate that, even in these challenging scenarios, our segmentation algorithm successfully identifies the garment region (see Figure 1 in https://github.com/hellohaia/img/blob/main/1.pdf).
>
> However, keypoint detection performs poorly under these conditions, as shown in Table 1.1. In the future, we will explore using depth images for both keypoint detection and garment segmentation under extreme conditions, which would mitigate the impact of color similarity.
>
> Table 1.1 Results of Segmentation and Keypoint Detection.
>
> | Method             | Precision (%)  |
> |--------------------|----------------|
> | Segmentation       | 10/10          |
> | Keypoint detection | 0/10           |
> |||
>
>
>
>
> ### 3. Discussion of End-to-End Approaches for Garment Folding
>
> Recent works, including end-to-end approaches like SpeedFolding [1] and UniFolding [2], have demonstrated competent garment folding capabilities. Below, we discuss their methods:
>
>
> SpeedFolding is a bimanual system that folds crumpled garments based on user-defined lines. While it performs well on short-sleeve garments, it doesn't generalize to other types. Additionally, it requires 4,300 human-annotated training samples, which is labor-intensive. Its performance degrades with highly crumpled garments, as key areas are occluded, and it struggles with controlling garment orientation during unfolding, limiting standardization.
>
>
> UniFolding uses the UFONet neural network to integrate unfolding and folding into a single policy. Tested on long- and short-sleeve shirts, it hasn't generalized to other garment types. Its performance degrades with highly crumpled garments, as keypoints can be obscured. It relies on labor-intensive human demonstrations in virtual reality for data collection. While it flattens garments, it cannot standardize them, and the model tends to focus on manipulating the sleeves, often causing the collar to roll in, preventing full flattening.
>
> Our work introduces dual-arm, multi-primitive policy to quickly unfold crumpled garments with standardization, improving downstream tasks like folding, ironing, and packing. Unlike their methods, our model trains without human-collected data in simulations and achieves zero-shot transfer, delivering strong real-world performance.
>
> ## References
>
> [1] Y. Avigal et al. Speedfolding: Learning efficient bimanual folding of garments. IROS 2022.
> [2] H. Xue et al. Unifolding: towards sample-efficient, scalable, and generalizable robotic garment folding. CoRL 2023.

---

> > ### Comment · Reviewer_fdEB · 2025-04-06
> >
> > Thank you for your response -- I will maintain my original acceptance score.

---

### Decision · Program_Chairs · 2025-05-01

**Decision:**

Accept (poster)

**Comment:**

This paper presents a practical approach to garment manipulation that combines unfolding and standardization in a unified framework.
The proposed approach employs a dual-arm, multi-primitive policy with dynamic fling to unfold and align garments.
The reviewers agree that this paper is well-written, and the proposed method is thoroughly evaluated in challenging real-world scenarios. The reviewers also find the novelty somewhat limited, but this is due to the nature of this work (a robotic/vision system). The area chair finds the paper very well-written and presented. The experiments are convincing.